# Gaussian-SLAM: Photo-realistic Dense SLAM with Gaussian Splatting

## Abstract

We present a dense simultaneous localization and mapping (SLAM) method that uses 3D Gaussians as a scene representation. Our approach enables interactive-time reconstruction and photo-realistic rendering from real-world single-camera RGBD videos. To this end, we propose a novel effective strategy for seeding new Gaussians for newly explored areas and their effective online optimization that is independent of the scene size and thus scalable to larger scenes. This is achieved by organizing the scene into sub-maps which are independently optimized and do not need to be kept in memory. We further accomplish frame-to-model camera tracking by minimizing photometric and geometric losses between the input and rendered frames. The Gaussian representation allows for high-quality photo-realistic real-time rendering of real-world scenes. Evaluation on synthetic and real-world datasets demonstrates competitive or superior performance in mapping, tracking, and rendering compared to existing neural dense SLAM methods.

## 1 Introduction

Simultaneous localization and mapping (SLAM) has been an active research topic for the past two decades Fuentes-Pacheco et al. (2015); Kazerouni et al. (2022). A major byproduct of that journey is the investigation of various scene representations to either push the tracking performance and mapping capabilities or to adapt it for more complex downstream tasks like path planning or semantic understanding. Specifically, earlier works focus on tracking using various scene representations like feature point clouds Klein & Murray (2007); Davison et al. (2007); Mur-Artal & Tardos (2017), surfels Whelan et al. (2015); Schops et al. (2019), depth maps Stühmer et al. (2010); Newcombe et al. (2011b), or implicit representations Newcombe et al. (2011a); Nießner et al. (2013); Dai et al. (2017b). Later works focus more on the map quality and density. With the advent of powerful neural scene representations like neural radiance fields Mildenhall et al. (2020) that allow for high fidelity view-synthesis, a rapidly growing body of dense neural SLAM methods Sucar et al. (2021); Huang et al. (2021); Zhu et al. (2022); Mahdi Johari et al. (2022); Tang et al. (2023); Wang et al. (2023); Sandström et al. (2023a); Zhang et al. (2023) has been developed. Despite their impressive gains in scene representation quality, these methods are still limited to small synthetic scenes and their re-rendering results are far from photo-realistic.

Recently, a novel scene representation based on Gaussian splatting Kerbl et al. (2023) has been shown to deliver on-par rendering performance with NeRFs while being an order of magnitude faster in rendering and optimization. Moreover, this scene representation is directly interpretable and can be directly manipulated which is desirable for many downstream tasks. With these advantages, the Gaussian splatting representation lends itself to be applied in an online SLAM system with real-time demands and opens the door to photo-realistic dense SLAM.

In this paper, we introduce Gaussian-SLAM, a dense RGBD SLAM system using 3D Gaussians to build a scene representation that allows for mapping, tracking, and photo-realistic re-rendering at interactive runtimes. An example of the high-fidelity rendering output of Gaussian-SLAM is depicted in Figure 1. In summary, our **contributions** include:

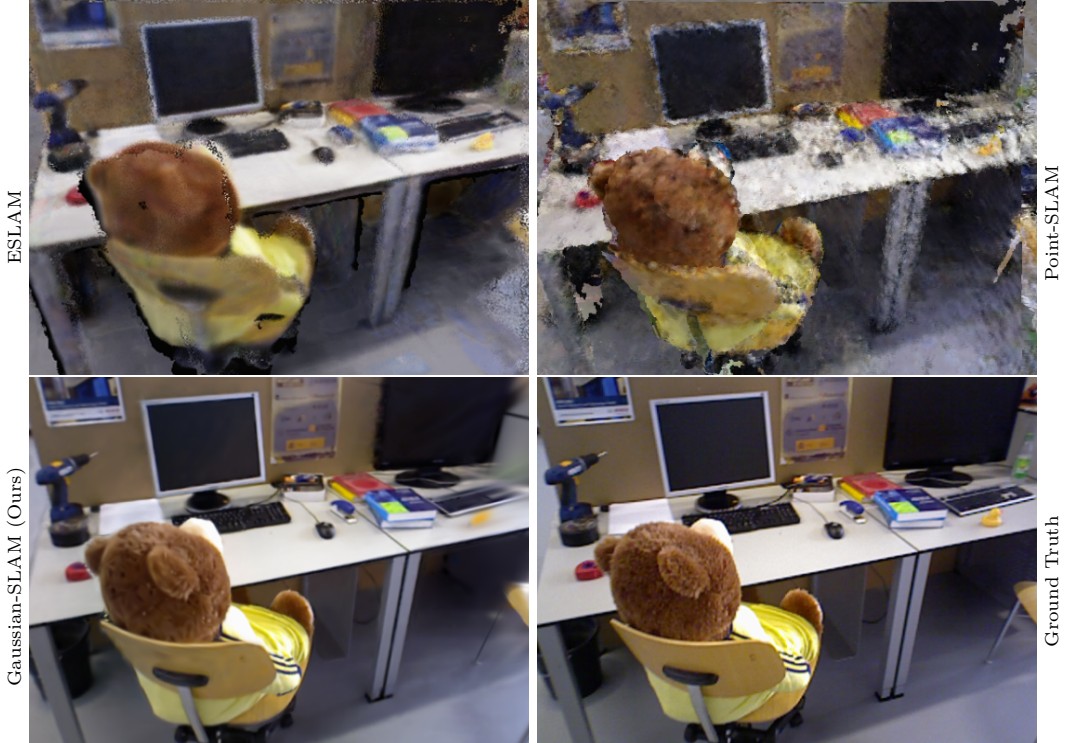

Figure 1: **Rendering results of Gaussian-SLAM.** Embedded into a dense SLAM pipeline, the 3D Gaussian-based scene representation allows for fast, photo-realistic rendering of scene views. This leads to high-quality rendering, especially on real-world data like this TUM-RGBD (Sturm et al., 2012) frame that contains many high-frequency details that other methods struggle to capture.

- A dense RGBD SLAM approach that uses 3D Gaussians to construct a scene representation allowing SOTA rendering results on real-world scenes.
- An extension of Gaussian splatting that better encodes geometry and allows reconstruction beyond radiance fields in a single-camera setup.
- An online optimization method for Gaussian splats that processes the map as sub-maps and introduces efficient seeding and optimization strategies.
- A frame-to-model tracker with the Gaussian splatting scene representation via photometric and geometric error minimization.

## 2 Related Work

**Dense Visual SLAM and Online Mapping.**

The seminal work of Curless and Levoy Curless & Levoy (1996) set the stage for a variety of 3D reconstruction methods using truncated signed distance functions (TSDF). A line of works was built upon it improving speed Newcombe et al. (2011a) through efficient implementation and volume integration, scalability through voxel hashing Nießner et al. (2013); Kähler et al. (2015); Oleynikova et al. (2017) and octree data structure Steinbrucker et al. (2013), as well as tracking with sparse image features Bylow et al. (2016) and loop closures Newcombe et al. (2011b); Schops et al. (2019); Cao et al. (2018); Dai et al. (2017b). Tackling the problem of unreliable depth maps, RoutedFusion Weder et al. (2020) introduced a learning-based fusion network for updating the TSDF in volumetric grids. This concept was further evolved by NeuralFusion Weder et al. (2021) and DI-Fusion Huang et al. (2021), which adopt implicit learning for scene representation, enhancing their robustness against outliers. Recent research has successfully achieved dense online recon-

struction using solely RGB cameras Murez et al. (2020); Choe et al. (2021); Božič et al. (2021); Stier et al. (2021); Sun et al. (2021); Sayed et al. (2022); Li et al. (2023), bypassing the need for depth data.

Recently, test-time optimization methods have become popular due to their ability to adapt to unseen scenes on the fly. Continuous Neural Mapping Yan et al. (2021), for instance, employs a continual mapping strategy from a series of depth maps to learn scene representation. Inspired by Neural Radiance Fields Mildenhall et al. (2020), there has been immense progress in dense surface reconstruction Oechsle et al. (2021); Wang et al. (2022) and accurate pose estimation Rosinol et al. (2022); Lin et al. (2021); Wang et al. (2021); Bian et al. (2022). These efforts have led to the development of comprehensive dense SLAM systems Yang et al. (2022a); Zhu et al. (2022; 2023); Sucar et al. (2021); Mahdi Johari et al. (2022); Zhang et al. (2023); Sandström et al. (2023b), showing a trend in the pursuit of precise and reliable visual SLAM. A comprehensive survey on online RGBD reconstruction can be found in Zollhöfer et al. (2018).

While the latest neural methods show impressive rendering capabilities on synthetic data, they struggle when applied to real-world data. Further, these methods are not yet practical for real-world applications due to computation requirements, slow speed, and the inability to effectively incorporate pose updates, as the neural representations rely on positional encoding. In contrast, our method shows impressive performance on real-world data, has competitive tracking and runtime, and uses a scene representation that naturally allows pose updates.

**Scene Representations for SLAM.** The majority of dense 3D scene representations for SLAM are grid-based, point-based, network-based, or hybrid. Among these, grid-based techniques are perhaps the most extensively researched. They further divide into methods using dense grids Zhu et al. (2022); Newcombe et al. (2011a); Weder et al. (2021; 2020); Curless & Levoy (1996); Sun et al. (2021); Božič et al. (2021); Li et al. (2022); Choi et al. (2015); Whelan et al. (2015); Zhou & Koltun (2013); Zhou et al. (2013); Whelan et al. (2012), hierarchical octrees Yang et al. (2022a); Steinbrucker et al. (2013); Marniok et al. (2017); Chen et al. (2013); Liu et al. (2023; 2020), voxel hashing Nießner et al. (2013); Kähler et al. (2015); Dai et al. (2017b); Wang et al. (2022); Müller et al. (2022), or distributed NeRFs Zhang et al. (2022); Tancik et al. (2022); Fang et al. (2023) for efficient memory management. Grids offer the advantage of simple and quick neighborhood lookups and context integration. However, a key limitation is the need to predefine grid resolution, which is not easily adjustable during reconstruction. This can result in inefficient memory usage in empty areas while failing to capture finer details due to resolution constraints.

Point-based approaches address some of the grid-related challenges and have been effectively utilized in 3D reconstruction Whelan et al. (2015); Schops et al. (2019); Cao et al. (2018); Chung et al. (2022); Kähler et al. (2015); Keller et al. (2013); Cho et al. (2021); Zhang et al. (2020). Unlike grid resolution, the density of points in these methods does not have to be predetermined and can naturally vary throughout the scene. Moreover, point sets can be efficiently concentrated around surfaces, not spending memory on modeling empty space. The trade-off for this adaptability is the complexity of finding neighboring points, as point sets lack structured connectivity. In dense SLAM, this challenge can be mitigated by transforming the 3D neighborhood search into a 2D problem via projection onto keyframes Whelan et al. (2015); Schops et al. (2019), or by organizing points within a grid structure for expedited searching Xu et al. (2022).

Network-based methods for dense 3D reconstruction provide a continuous scene representation by implicitly modeling it with coordinate-based networks Azinović et al. (2022); Sucar et al. (2021); Wang et al. (2022); Ortiz et al. (2022); Yan et al. (2021); Yang et al. (2022b); Mescheder et al. (2019); Li et al. (2023); Zhang et al. (2023); Wang et al. (2023). This representation can capture high-quality maps and textures. However, they are generally unsuitable for online scene reconstruction due to their inability to update local scene regions and to scale for larger scenes.

Outside these three primary categories, some studies have explored alternative representations like surfels Mihajlovic et al. (2021); Gao et al. (2023) and neural planes Mahdi Johari et al. (2022); Peng et al. (2020). Parameterized surface elements are generally not great at modeling a flexible shape template while feature planes struggle with scene reconstructions containing multiple surfaces, due to their overly compressed representation. Recently, Kerbl et al. (2023) proposed to represent a scene with 3D Gaussians. The Gaussian parameters are optimized via differential rendering with multi-view supervision. While being

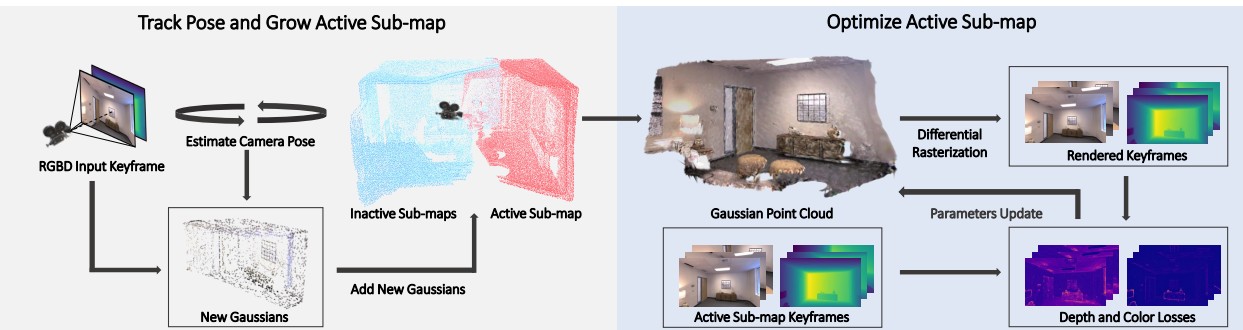

Figure 2: **Gaussian-SLAM architecture.** For every input keyframe the camera pose is estimated using depth and color losses against the *active* sub-map. Given an estimated pose, the RGBD frame is transformed into 3D and subsampled based on color gradient and the rendered alpha mask. Points from the subsampled point clouds located in low-density areas of the *active* sub-map are used to initialize new 3D Gaussians. These sparse 3D Gaussians are then added to the Gaussian point cloud of the *active* sub-map and are jointly optimized with the depth maps and color images from all contributing keyframes of this sub-map.

very efficient and achieving impressive rendering results, this representation is tailored for fully-observed multi-view environments and does not encode geometry well.

Recently, several methods Keetha et al. (2023); Matsuki et al. (2023a); Yan et al. (2024); Huang et al. (2023) have used Gaussian Splatting Kerbl et al. (2023) for SLAM. While most splatting-based methods use gradient-based map densification similar to Kerbl et al. (2023), we follow a more controlled approach with exact thresholding by utilizing fast nearest-neighbor search and alpha masking. Further, unlike others, our mapping pipeline does not require holding all the 3D Gaussians in the GPU memory, allowing our method to scale and not slow down as more areas are covered. Moreover, while in other works the 3D Gaussians are very densely seeded, our color gradient and masking-based seeding strategy allows for sparser seeding while preserving SOTA rendering quality. Finally, in contrast to Matsuki et al. (2023a); Yan et al. (2024); Huang et al. (2023), our tracking does not rely on explicitly computed camera pose derivatives and is implemented in PyTorch.

## 3  Method

The key idea of our approach is to construct a map using 3D Gaussians (Kerbl et al., 2023) as a main building block to make single-camera RGBD neural SLAM scalable, faster and achieve better rendering on real-world datasets. We introduce a novel efficient mapping process with bounded computational cost in a sequential single-camera setup, a challenging scenario for traditional 3D Gaussian Splatting. To enable traditional Gaussian splats to render accurate geometry we extend them by adding a differential depth rendering, explicitly computing gradients for the Gaussian parameters updates. Finally, we develop a novel frame-to-model tracking approach relying on our 3D map representation. Figure 2 provides an overview of our method. We now explain our pipeline, starting with an overview of classical Gaussian splatting, and continuing with map construction and optimization, geometry encoding, and tracking.

### 3.1  Gaussian Splatting

Gaussian splatting (Kerbl et al., 2023) is an effective method for representing 3D scenes with novel-view synthesis capability. This approach is notable for its speed, without compromising the rendering quality. In Kerbl et al. (2023), 3D Gaussians are initialized from a sparse Structure-from-Motion point cloud of a scene. With images observing the scene from different angles, the Gaussian parameters are optimized using differentiable rendering. During training, 3D Gaussians are adaptively added or removed to better render the images based on a set of heuristics.

A single 3D Gaussian is parameterized by mean $\mu \in \mathbb{R}^3$, covariance $\Sigma \in \mathbb{R}^{3 \times 3}$, opacity $o \in \mathbb{R}$, and RGB color $C \in \mathbb{R}^3$. The mean of a projected (splatted) 3D Gaussian in the 2D image plane $\mu^I$ is computed as follows:

$$\mu^I = \pi\big(P(T_{wc}\mu_{\text{homogeneous}})\big) \; , \tag{1}$$

where $T_{wc} \in SE(3)$ is the world-to-camera transformation, $P \in \mathbb{R}^{4 \times 4}$ is an OpenGL-style projection matrix, $\pi : \mathbb{R}^4 \to \mathbb{R}^2$ is a projection to pixel coordinates. The 2D covariance $\Sigma^I$ of a splatted Gaussian is computed as:

$$\Sigma^I = JR_{wc}\Sigma R_{wc}^T J^T \; , \tag{2}$$

where $J \in \mathbb{R}^{2 \times 3}$ is an affine transformation from Zwicker et al. (2001), $R_{wc} \in SO(3)$ is the rotation component of world-to-camera transformation $T_{wc}$. We refer to Zwicker et al. (2001) for further details about the projection matrices. Color $C$ along one channel $ch$ at a pixel $i$ influenced by $m$ ordered Gaussians is rendered as:

$$C_i^{ch} = \sum_{j \leq m} C_j^{ch} \cdot \alpha_j \cdot Tr_j \; , \; \text{with} \; Tr_j = \prod_{k < j}(1 - \alpha_k) \; , \tag{3}$$

with $\alpha_j$ is computed as:

$$\alpha_j = o_j \cdot \exp(-\sigma_j) \quad \text{and} \quad \sigma_j = \frac{1}{2}\Delta_j^T \Sigma_j^{I-1} \Delta_j \; , \tag{4}$$

where $\Delta_j \in \mathbb{R}^2$ is the offset between the pixel coordinates and the 2D mean of a splatted Gaussian. The parameters of the 3D Gaussians are iteratively optimized by minimizing the photometric loss between rendered and training images. During optimization, $C$ is encoded with spherical harmonics $SH \in \mathbb{R}^{15}$ to account for direction-based color variations. Covariance is decomposed as $\Sigma = RSS^T R^T$, where $R \in \mathbb{R}^{3 \times 3}$ and $S = \text{diag}(s) \in \mathbb{R}^{3 \times 3}$ are rotation and scale respectively to preserve covariance positive semi-definite property during gradient-based optimization.

### 3.2 3D Gaussian-based Map

To avoid catastrophic forgetting and overfitting and make the mapping computationally feasible in a single-camera stream scenario we process the input in chunks (sub-maps). Every sub-map covers several keyframes observing it and is represented with a separate 3D Gaussian point cloud. Formally, we define a sub-map Gaussian point cloud $P^s$ as a collection of $N$ 3D Gaussians:

$$P^s = \{G(\mu_i^s, \Sigma_i^s, o_i^s, C_i^s) \,|\, i = 1, \ldots, N\} \; . \tag{5}$$

**Sub-map Initialization.** A sub-map starts with the first frame and grows incrementally with newly incoming keyframes. As the explored area grows, a new sub-map is needed to cover the unseen regions and avoid storing all the Gaussians in GPU memory. Instead of using a fixed interval when creating a new sub-map (Choi et al., 2015; Dai et al., 2017b; Maier et al., 2014), an initialization strategy that relies on the camera motion (Cao et al., 2018; Stückler & Behnke, 2014) is used. Specifically, a new sub-map is created when the current frame's estimated translation relative to the first frame of the active sub-map exceeds a predefined threshold, $d_{\text{thre}}$, or when the estimated Euler angle surpasses $\theta_{\text{thre}}$. At any time, only active sub-map is processed. This approach bounds the compute cost and ensures that optimization remains fast while exploring larger scenes.

**Sub-map Building.** Every new keyframe may add 3D Gaussians to the active sub-map to account for the newly observed parts of the scene. Following the pose estimation for the current keyframe, a dense point cloud is computed from keyframe RGBD measurements. At the beginning of each sub-map, we sample $M_u$ uniformly and $M_c$ points from the keyframe point cloud in high color gradient regions to add new Gaussians. For the following keyframes of the sub-map, we sample $M_k$ points uniformly from the regions with the rendered alpha values lower than a threshold $\alpha_n$. This allows for growing the map in areas sparsely covered by the 3D Gaussians. New Gaussians are added to the sub-map using sampled points that have no neighbors within a search radius $\rho$ in the current sub-map. The new Gaussians are anisotropic and their scales are defined based on the nearest neighbor distance within the active sub-map. This densification

strategy substantially differs from Kerbl et al. (2023) where new Gaussians were added and pruned based on the gradient values during optimization and gives fine-grained control over the number of Gaussians.

**Sub-map Optimization.** All Gaussians in the active sub-map are jointly optimized every time new Gaussians are added to the sub-map for a fixed number of iterations minimizing the loss equation 12. We do not clone or prune the Gaussians as done in Kerbl et al. (2023) during optimization to preserve geometry density obtained from the depth sensor, decrease computation time, and better control the number of Gaussians. We optimize the active sub-map to render the depth and color of all its keyframes. We directly optimize RGB color without using spherical harmonics to speed up optimization and improve tracking performance. See supplementary material for more details. In Gaussian splatting (Kerbl et al., 2023) the scene representation is optimized for many iterations over all the training views. However, this approach does not suit the SLAM setup where speed is crucial. Naively optimizing with an equal number of iterations for all keyframes results in underfitting or excessive time spent on optimization. We solve this by optimizing only the keyframes in the active sub-map and spending at least 40% of iterations on the new keyframe.

### 3.3 Geometry and Color Encoding

While Gaussian Splatting (Kerbl et al., 2023) is good at rendering images, the rendered depth maps are of limited accuracy since there is no direct depth supervision. We tackle this problem with an additional depth loss. To render the depth $D_i$ at pixel $i$ that is influenced by $m$ ordered Gaussians we compute:

$$D_i = \sum_{j \leq m} \mu_j^z \cdot \alpha_j \cdot T_j \ , \tag{6}$$

where $\mu_j^z$ is a $z$ component of the mean of a 3D Gaussian, $\alpha_j$ and $T_j$ are the same as in equation 3. To update the 3D Gaussian parameters based on the observed depth, we derive the gradients of the depth loss w.r.t. the 3D Gaussians' means, covariances, and opacity. Denoting the depth loss as $L_{\text{depth}}$, we follow the chain rule to compute the gradient for the mean update of the Gaussian $j$:

$$\frac{\partial L_{\text{depth}}}{\partial \mu_j} = \frac{\partial L_{\text{depth}}}{\partial D_i} \frac{\partial D_i}{\partial \alpha_j} \frac{\partial \alpha_j}{\partial \mu_j} \ , \tag{7}$$

where $\frac{\partial L_{\text{depth}}}{\partial D_i}$ is computed with PyTorch autograd using equation 9 and $\frac{\partial \alpha_j}{\partial \mu_j}$ is derived as in Kerbl et al. (2023). We derive $\frac{\partial D_i}{\partial \alpha_j}$ as:

$$\frac{\partial D_i}{\partial \alpha_j} = \mu_j^z \cdot T_j - \frac{\sum_{u>j} \mu_u^z \alpha_u T_u}{1 - \alpha_j} \ . \tag{8}$$

The gradients for covariance and opacity are computed similarly. Apart from $\frac{\partial L_{\text{depth}}}{\partial D_i}$, all gradients are explicitly computed in CUDA to preserve the optimization speed of the unified rendering pipeline. For depth supervision, we use the loss:

$$L_{\text{depth}} = |\hat{D} - D|_1 \ , \tag{9}$$

with $D$ and $\hat{D}$ being the ground-truth and reconstructed depth maps, respectively. For the color supervision we use a weighted combination of $L_1$ and SSIM (Wang et al., 2004) losses:

$$L_{\text{color}} = (1 - \lambda) \cdot |\hat{I} - I|_1 + \lambda \big(1 - \text{SSIM}(\hat{I}, I)\big) \ , \tag{10}$$

where $I$ is the original image, $\hat{I}$ is the rendered image, and $\lambda = 0.2$.

When seeded sparsely as in our case, a few 3D Gaussians sometimes elongate too much in scale. To overcome this, we add an isotropic regularization term $L_{\text{reg}}$ when optimizing a sub-map $K$:

$$L_{\text{reg}} = \frac{\sum_{k \in K} |s_k - \overline{s}_k|_1}{|K|}, \tag{11}$$

where $s_k \in \mathbb{R}^3$ is the scale of a 3D Gaussian, $\overline{s}_k$ is the mean sub-map scale, and $|K|$ is the number of Gaussians in the sub-map. Finally, we optimize color, depth, and regularization terms together:

$$L = \lambda_{\text{color}} \cdot L_{\text{color}} + \lambda_{\text{depth}} \cdot L_{\text{depth}} + \lambda_{\text{reg}} \cdot L_{\text{reg}} \ , \tag{12}$$

where $\lambda_{\text{color}}, \lambda_{\text{depth}}, \lambda_{\text{reg}} \cdot \in \mathbb{R}_{\geq 0}$ are weights for the corresponding losses.

### 3.4 Tracking

We perform frame-to-model tracking based on the mapped scene. We initialize the current camera pose $T_i$ with a constant speed assumption:

$$T_i = T_{i-1} \cdot T_{i-2}^{-1} \cdot T_{i-1} \ , \tag{13}$$

where pose $T_i \in SE(3)$ is a transformation matrix. To estimate the camera pose we minimize the tracking loss $L_{\text{tracking}}$ with respect to relative camera pose $T_{i-1,i}$ between frames $i-1$ and $i$ as follows:

$$\underset{T_{i-1,i}}{\arg\min} \, L_{\text{tracking}}\Big(\hat{I}(T_{i-1,i}), \hat{D}(T_{i-1,i}), I_i, D_i, \alpha\Big) \ , \tag{14}$$

where $\hat{I}(T_{i-1,i})$ and $\hat{D}(T_{i-1,i})$ are the rendered color and depth from the sub-map transformed with the relative transformation $T_{i-1,i}$, $C_i$ and $D_i$ are the input color and depth map at frame $i$.

We introduce soft alpha and error masking to not contaminate the tracking loss with the pixels from previously unobserved or poorly reconstructed areas. Soft alpha mask $M_{\text{alpha}}$ is a polynomial of the alpha map rendered directly from the active sub-map. Error boolean mask $M_{\text{inlier}}$ discards all the pixels where the color and depth errors are larger than a frame-relative error threshold:

$$L_{\text{tracking}} = \sum M_{\text{inlier}} \cdot M_{\text{alpha}} \cdot (\lambda_c |\hat{I} - I|_1 + (1 - \lambda_c)|\hat{D} - D|_1). \tag{15}$$

The weighting ensures the optimization is guided by well-reconstructed regions where the accumulated alpha values are close to 1 and rendering quality is high. During optimization, all the 3D Gaussian parameters are frozen.

## 4 Experiments

We first describe our experimental setup and then evaluate our method against state-of-the-art dense neural RGBD SLAM methods on synthetic (Straub et al., 2019) and real-world datasets (Sturm et al., 2012; Dai et al., 2017a; Yeshwanth et al., 2023). The reported results are the average of 3 runs using different seeds. The tables highlight best results as first , second , third .

**Implementation Details.** We set the number of uniformly sampled points $M_u = 600000$ for Replica (Straub et al., 2019), 100000 for TUM-RGBD (Sturm et al., 2012) and ScanNet (Dai et al., 2017a), and 400000 for ScanNet++ (Yeshwanth et al., 2023). Number of points sampled in high-gradient regions $M_c$ is set to 50000 for all datasets. For the first keyframe in a sub-map, the number of mapping iterations is set to 1,000 for Replica, 100 for TUM-RGBD and ScanNet, and 500 for ScanNet++. For the subsequent keyframes in a sub-map, the iteration count is set to 100 across all datasets. Every 5th frame is considered as a keyframe for all the datasets. When selecting point candidates from subsequent keyframes, we use alpha threshold $\alpha_n = 0.6$. We use FAISS (Johnson et al., 2019) GPU implementation to find nearest neighbors when choosing point candidates to add as new Gaussians and set the search radius $\rho = 0.01\,m$ for all the datasets. For new sub-map initialization, we set $d_{\text{thre}} = 0.5\,m$ and $\theta_{\text{thre}} = 50°$. For sub-map optimization, the best results were obtained with $\lambda_{\text{color}}$, $\lambda_{\text{reg}}$ and $\lambda_{\text{depth}}$ to 1. We spend at least 40% mapping iterations on the newly added keyframe during sub-map optimization. To mesh the scene, we render depth and color every fifth frame over the estimated trajectory and use TSDF Fusion Curless & Levoy (1996) with voxel size $1\,cm$ similar to Sandström et al. (2023a). Further details are provided in the supplement.

**Datasets.** The Replica dataset (Straub et al., 2019) comprises high-quality 3D reconstructions of a variety of indoor scenes. We utilize the publicly available dataset collected by Sucar et al. (2021), which provides trajectories from an RGBD sensor. Further, we demonstrate that our framework achieves SOTA results on real-world data by using the TUM-RGBD (Sturm et al., 2012), ScanNet (Dai et al., 2017a) and ScanNet++ (Yeshwanth et al., 2023) datasets. The poses for TUM-RGBD were captured using an external motion capture system while ScanNet uses poses estimated by BundleFusion (Dai et al., 2017b), and ScanNet++ obtains poses by registering the images with a laser scan. Since ScanNet++ is not specifically designed for benchmarking neural SLAM, it has larger camera movements. Therefore, we choose 5 scenes where the first 250 frames are smooth in trajectory and use them for benchmarking.

**Evaluation Metrics.** To assess tracking accuracy, we use ATE RMSE (Sturm et al., 2012), and for rendering we compute PSNR, SSIM (Wang et al., 2004) and LPIPS (Zhang et al., 2018). All rendering metrics are evaluated by rendering full-resolution images along the estimated trajectory with mapping intervals similar to Sandström et al. (2023a). We also follow Sandström et al. (2023a) to measure reconstruction performance on meshes produced by marching cubes (Lorensen & Cline, 1987). The reconstructions are also evaluated using the F1-score - the harmonic mean of the Precision (P) and Recall (R). We use a distance threshold of 1 cm for all evaluations. We further provide the depth L1 metric for unseen views as in Zhu et al. (2022).

**Baseline Methods.** We primarily compare our method to existing state-of-the-art dense neural RGBD SLAM methods such as NICE-SLAM (Zhu et al., 2022), Vox-Fusion (Yang et al., 2022a), ESLAM (Mahdi Johari et al., 2022), and Point-SLAM (Sandström et al., 2023a), as well as the 3DGS-based methods SplaTAM (Keetha et al., 2023) and MonoGS Matsuki et al. (2023a).

**Rendering Performance.** Table 1 compares rendering performance and shows improvements over all the existing dense neural RGBD SLAM methods on synthetic data. Table 2 and Table 3 show our state-of-the-art rendering performance on real-world datasets. Figure 3 shows exemplary full-resolution renderings where Gaussian-SLAM yields more accurate details. Qualitative results on novel views are provided as a video in the supplementary.

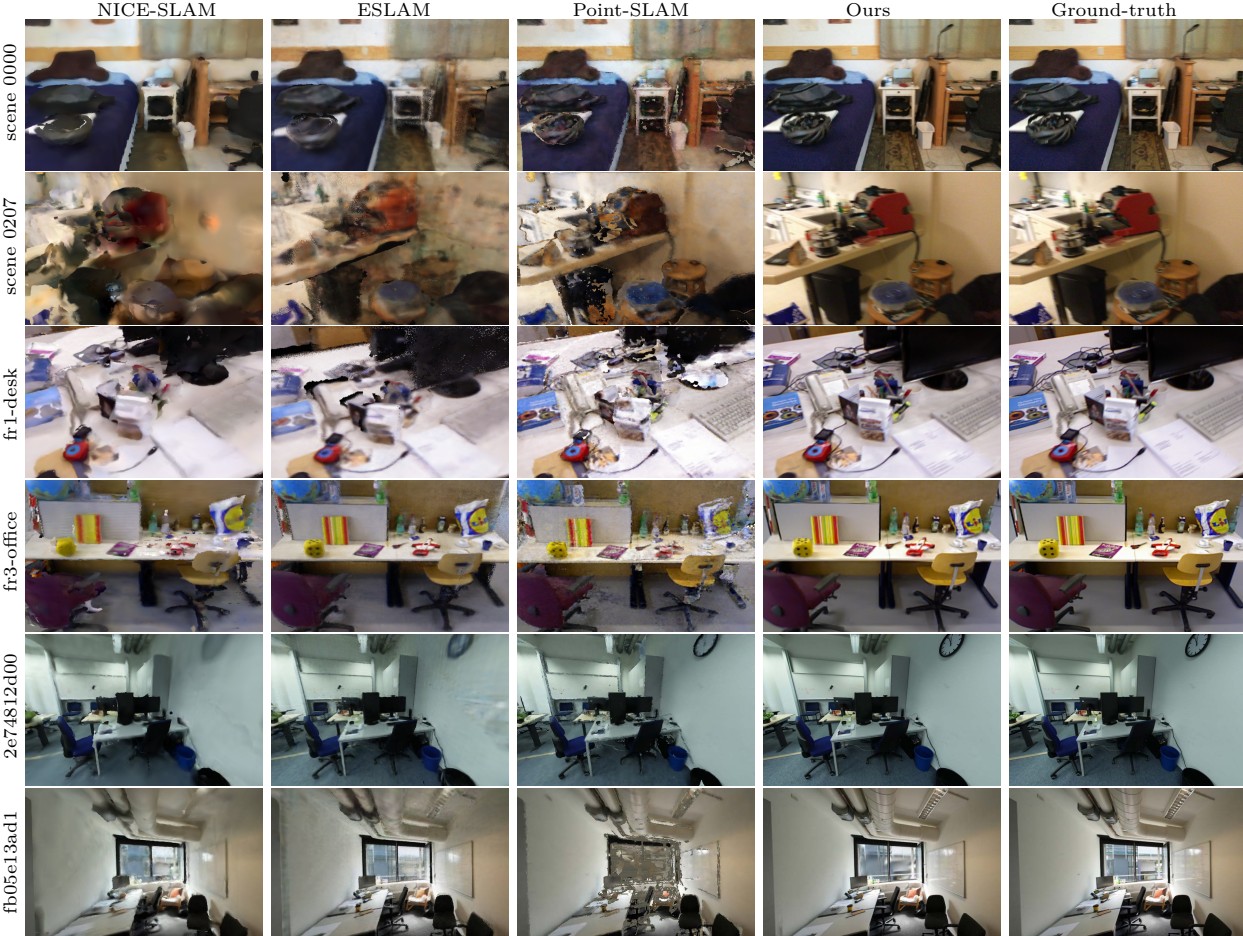

Figure 3: **Rendering performance on ScanNet, TUM-RGBD and ScanNet++ dataset**. Thanks to 3D Gaussian splatting, Gaussian-SLAM can encode more high-frequency details and substantially increase the quality of the renderings (please zoom in for a better view of the details). This is also supported by the quantitative results in Table 2 and 3.

Table 1: **Rendering performance on Replica dataset.** We outperform all existing dense neural RGBD methods on the commonly reported rendering metrics.

| Method | Metric | Rm0 | Rm1 | Rm2 | Off0 | Off1 | Off2 | Off3 | Off4 | Avg. |
|---|---|---|---|---|---|---|---|---|---|---|
| NICE-SLAM (Zhu et al., 2022) | PSNR↑ | 22.12 | 22.47 | 24.52 | 29.07 | 30.34 | 19.66 | 22.23 | 24.94 | 24.42 |
| | SSIM↑ | 0.689 | 0.757 | 0.814 | 0.874 | 0.886 | 0.797 | 0.801 | 0.856 | 0.809 |
| | LPIPS↓ | 0.330 | 0.271 | 0.208 | 0.229 | 0.181 | 0.235 | 0.209 | 0.198 | 0.233 |
| Vox-Fusion (Yang et al., 2022a) | PSNR↑ | 22.39 | 22.36 | 23.92 | 27.79 | 29.83 | 20.33 | 23.47 | 25.21 | 24.41 |
| | SSIM↑ | 0.683 | 0.751 | 0.798 | 0.857 | 0.876 | 0.794 | 0.803 | 0.847 | 0.801 |
| | LPIPS↓ | 0.303 | 0.269 | 0.234 | 0.241 | 0.184 | 0.243 | 0.213 | 0.199 | 0.236 |
| ESLAM (Mahdi Johari et al., 2022) | PSNR↑ | 25.25 | 27.39 | 28.09 | 30.33 | 27.04 | 27.99 | 29.27 | 29.15 | 28.06 |
| | SSIM↑ | 0.874 | 0.89 | 0.935 | 0.934 | 0.910 | 0.942 | 0.953 | 0.948 | 0.923 |
| | LPIPS↓ | 0.315 | 0.296 | 0.245 | 0.213 | 0.254 | 0.238 | 0.186 | 0.210 | 0.245 |
| Point-SLAM (Sandström et al., 2023a) | PSNR↑ | 32.40 | 34.08 | 35.50 | 38.26 | 39.16 | 33.99 | 33.48 | 33.49 | 35.17 |
| | SSIM↑ | 0.974 | 0.977 | 0.982 | 0.983 | 0.986 | 0.960 | 0.960 | 0.979 | 0.975 |
| | LPIPS↓ | 0.113 | 0.116 | 0.111 | 0.100 | 0.118 | 0.156 | 0.132 | 0.142 | 0.124 |
| SplaTAM(Keetha et al., 2023) | PSNR↑ | 32.86 | 33.89 | 35.25 | 38.26 | 39.17 | 31.97 | 29.70 | 31.81 | 34.11 |
| | SSIM↑ | 0.98 | 0.97 | 0.98 | 0.98 | 0.98 | 0.97 | 0.95 | 0.95 | 0.97 |
| | LPIPS↓ | 0.07 | 0.10 | 0.08 | 0.09 | 0.09 | 0.10 | 0.12 | 0.15 | 0.10 |
| MonoGS (Matsuki et al., 2023b) | PSNR↑ | 35.99 | 38.15 | 38.85 | 42.85 | 43.42 | 37.31 | 37.27 | 37.25 | 38.89 |
| | SSIM↑ | 0.96 | 0.967 | 0.971 | 0.982 | 0.981 | 0.969 | 0.968 | 0.964 | 0.970 |
| | LPIPS↓ | 0.048 | 0.052 | 0.051 | 0.029 | 0.032 | 0.044 | 0.041 | 0.062 | 0.045 |
| Gaussian-SLAM (ours) | PSNR↑ | 38.88 | 41.80 | 42.44 | 46.40 | 45.29 | 40.10 | 39.06 | 42.65 | 42.08 |
| | SSIM↑ | 0.993 | 0.996 | 0.996 | 0.998 | 0.997 | 0.997 | 0.997 | 0.997 | 0.996 |
| | LPIPS↓ | 0.017 | 0.018 | 0.019 | 0.015 | 0.016 | 0.020 | 0.020 | 0.020 | 0.018 |

Table 2: **Rendering performance on TUM-RGBD dataset.** We outperform existing dense neural RGBD methods on the commonly reported rendering metrics. For qualitative results, see Figure 3.

| Method | Metric | fr1/desk | fr2/xyz | fr3/office | Avg. |
|---|---|---|---|---|---|
| NICE-SLAM (Zhu et al., 2022) | PSNR↑ | 13.83 | 17.87 | 12.890 | 14.86 |
| | SSIM↑ | 0.569 | 0.718 | 0.554 | 0.614 |
| | LPIPS↓ | 0.482 | 0.344 | 0.498 | 0.441 |
| Vox-Fusion (Yang et al., 2022a) | PSNR↑ | 15.79 | 16.32 | 17.27 | 16.46 |
| | SSIM↑ | 0.647 | 0.706 | 0.677 | 0.677 |
| | LPIPS↓ | 0.523 | 0.433 | 0.456 | 0.471 |
| ESLAM (Mahdi Johari et al., 2022) | PSNR↑ | 11.29 | 17.46 | 17.02 | 15.26 |
| | SSIM↑ | 0.666 | 0.310 | 0.457 | 0.478 |
| | LPIPS↓ | 0.358 | 0.698 | 0.652 | 0.569 |
| Point-SLAM (Sandström et al., 2023a) | PSNR↑ | 13.87 | 17.56 | 18.43 | 16.62 |
| | SSIM↑ | 0.627 | 0.708 | 0.754 | 0.696 |
| | LPIPS↓ | 0.544 | 0.585 | 0.448 | 0.526 |
| SplaTAM (Keetha et al., 2023) | PSNR↑ | 22.00 | 24.50 | 21.90 | 22.80 |
| | SSIM↑ | 0.857 | 0.947 | 0.876 | 0.893 |
| | LPIPS↓ | 0.232 | 0.100 | 0.202 | 0.178 |
| MonoGS (Matsuki et al., 2023b) | PSNR↑ | 23.64 | 24.81 | 24.96 | 24.47 |
| | SSIM↑ | 0.783 | 0.795 | 0.839 | 0.806 |
| | LPIPS↓ | 0.248 | 0.222 | 0.207 | 0.226 |
| Gaussian-SLAM (ours) | PSNR↑ | 24.01 | 25.02 | 26.13 | 25.05 |
| | SSIM↑ | 0.924 | 0.924 | 0.939 | 0.929 |
| | LPIPS↓ | 0.178 | 0.186 | 0.141 | 0.168 |

**Novel View Synthesis.** In Table 4 we report the novel view synthesis results on the selected Scannet++ (Yeshwanth et al., 2023) scenes. The evaluated novel views in this dataset are not sampled from the input stream, but held-out views, which can better assess the extrapolation capability of the method. Gaussian-SLAM demonstrates clear advantage and outperforms the second-best (Keetha et al., 2023) by an average of 3.6 dB in PSNR. This result indicates that our method avoids overfitting on the training views, achieving strong rendering performance without compromising novel view synthesis capability.

**Tracking Performance.** In Table 5, Table 6, Table 8 and Table 9 we report the tracking accuracy on one synthetic (Straub et al., 2019) and three real-world datasets. Our method outperforms the nearest competitor by 14% on Replica. On TUM-RGBD dataset (Sturm et al., 2012), Gaussian-SLAM also performs better than all baseline methods. On ScanNet dataset, our method exhibits a drift due to low-quality depth maps

Table 3: **Rendering performance on ScanNet dataset.** We outperform existing dense neural RGBD methods on the commonly reported rendering metrics by a significant margin. For qualitative results, see Figure 3.

| Method | Metric | 0000 | 0059 | 0106 | 0169 | 0181 | 0207 | Avg. |
|---|---|---|---|---|---|---|---|---|
| NICE-SLAM (Zhu et al., 2022) | PSNR↑ | 18.71 | 16.55 | 17.29 | 18.75 | 15.56 | 18.38 | 17.54 |
| | SSIM↑ | 0.641 | 0.605 | 0.646 | 0.629 | 0.562 | 0.646 | 0.621 |
| | LPIPS↓ | 0.561 | 0.534 | 0.510 | 0.534 | 0.602 | 0.552 | 0.548 |
| Vox-Fusion (Yang et al., 2022a) | PSNR↑ | 19.06 | 16.38 | 18.46 | 18.69 | 16.75 | 19.66 | 18.17 |
| | SSIM↑ | 0.662 | 0.615 | 0.753 | 0.650 | 0.666 | 0.696 | 0.673 |
| | LPIPS↓ | 0.515 | 0.528 | 0.439 | 0.513 | 0.532 | 0.500 | 0.504 |
| ESLAM (Mahdi Johari et al., 2022) | PSNR↑ | 15.70 | 14.48 | 15.44 | 14.56 | 14.22 | 17.32 | 15.29 |
| | SSIM↑ | 0.687 | 0.632 | 0.628 | 0.656 | 0.696 | 0.653 | 0.658 |
| | LPIPS↓ | 0.449 | 0.450 | 0.529 | 0.486 | 0.482 | 0.534 | 0.488 |
| Point-SLAM (Sandström et al., 2023a) | PSNR↑ | 21.30 | 19.48 | 16.80 | 18.53 | 22.27 | 20.56 | 19.82 |
| | SSIM↑ | 0.806 | 0.765 | 0.676 | 0.686 | 0.823 | 0.750 | 0.751 |
| | LPIPS↓ | 0.485 | 0.499 | 0.544 | 0.542 | 0.471 | 0.544 | 0.514 |
| SplaTAM (Keetha et al., 2023) | PSNR↑ | 19.33 | 19.27 | 17.73 | 21.97 | 16.76 | 19.8 | 19.14 |
| | SSIM↑ | 0.660 | 0.792 | 0.690 | 0.776 | 0.683 | 0.696 | 0.716 |
| | LPIPS↓ | 0.438 | 0.289 | 0.376 | 0.281 | 0.420 | 0.341 | 0.358 |
| MonoGS Matsuki et al. (2023b) | PSNR↑ | 23.77 | 22.539 | 24.94 | 28.47 | **28.23** | 26.79 | 25.79 |
| | SSIM↑ | 0.772 | 0.793 | 0.863 | 0.850 | 0.897 | 0.829 | 0.834 |
| | LPIPS↓ | 0.435 | 0.308 | 0.266 | 0.273 | 0.247 | 0.336 | 0.311 |
| Gaussian-SLAM(ours) | PSNR↑ | **28.54** | **26.21** | **26.26** | **28.60** | 27.79 | **28.63** | **27.67** |
| | SSIM↑ | **0.926** | **0.934** | **0.926** | **0.917** | **0.922** | **0.914** | **0.923** |
| | LPIPS↓ | **0.271** | **0.211** | **0.217** | **0.226** | 0.277 | **0.288** | **0.248** |

Table 4: **Novel view synthesis performance on ScanNet++ dataset** (PSNR ↑ [dB]). Gaussian-SLAM demonstrates a clear advantage, outperforming the second-best Keetha et al. (2023) by an average of 3.6 dB on held-out views. Our calculation includes all pixels, regardless of whether they have valid depth input.

| Method | b20a261fdf | 8b5caf3398 | fb05e13ad1 | 2e74812d00 | 281bc17764 | Average |
|---|---|---|---|---|---|---|
| ESLAM Mahdi Johari et al. (2022) | 13.63 | 11.86 | 11.83 | 10.59 | 10.64 | 11.71 |
| SplaTAM Keetha et al. (2023) | 23.95 | 22.66 | 13.95 | 8.47 | 20.06 | 17.82 |
| MonoGS Matsuki et al. (2023a) | 15.41 | 14.04 | 12.07 | 10.72 | 12.36 | 12.92 |
| Gaussian-SLAM | **25.92** | **24.49** | **16.36** | **18.56** | **22.04** | **21.47** |

and a large amount of motion blur. On ScanNet++, where the camera motions are much larger compared to other datasets, our Gaussian splatting-based method performs significantly better than state-of-the-art NeRF-based methods, demonstrating greater robustness. What's more, Table 7 reports the tracking results on the large `apartment0` scene from (Bruns et al., 2024), the top performing methods fail due to tracking failure or out-of-GPU-memory on a 24GB memory GPU, while ours perform consistently well.

Table 5: **Tracking performance on Replica dataset** (ATE RMSE ↓ [cm]). We outperform all other methods in on Replica.

| Method | Rm0 | Rm1 | Rm2 | Off0 | Off1 | Off2 | Off3 | Off4 | Avg. |
|---|---|---|---|---|---|---|---|---|---|
| NICE-SLAM (Zhu et al., 2022) | 1.69 | 2.04 | 1.55 | 0.99 | 0.90 | 1.39 | 3.97 | 3.08 | 1.95 |
| Vox-Fusion (Yang et al., 2022a) | **0.27** | 1.33 | 0.47 | 0.70 | 1.11 | 0.46 | 0.26 | 0.58 | 0.65 |
| ESLAM (Mahdi Johari et al., 2022) | 0.71 | 0.70 | 0.52 | 0.57 | 0.55 | 0.58 | 0.72 | 0.63 | 0.63 |
| Point-SLAM (Sandström et al., 2023a) | 0.61 | 0.41 | 0.37 | 0.38 | 0.48 | 0.54 | 0.72 | 0.63 | 0.52 |
| SplaTAM(Keetha et al., 2023) | 0.31 | 0.40 | 0.29 | 0.47 | 0.27 | 0.29 | 0.32 | 0.55 | 0.36 |
| MonoGS (Matsuki et al., 2023b) | 0.33 | **0.22** | 0.29 | **0.36** | **0.19** | **0.25** | **0.12** | 0.81 | 0.32 |
| Gaussian SLAM (ours) | 0.29 | 0.29 | **0.22** | 0.37 | 0.23 | 0.41 | 0.30 | **0.35** | **0.31** |

**Reconstruction Performance.** In Table 10 we compare our method to NICE-SLAM (Zhu et al., 2022), Vox-Fusion (Yang et al., 2022a), ESLAM (Mahdi Johari et al., 2022), Point-SLAM (Sandström et al., 2023a), and SplaTAM (Keetha et al., 2023) in terms of the geometric reconstruction accuracy on the Replica dataset.

Table 6: **Tracking performance on TUM-RGBD dataset** (ATE RMSE↓ [cm]).

| Method | desk | xyz | office | Avg. |
|---|---|---|---|---|
| NICE-SLAM (Zhu et al., 2022) | 4.3 | 31.7 | 3.9 | 13.3 |
| Vox-Fusion (Yang et al., 2022a) | 3.5 | 1.5 | 26.0 | 10.3 |
| ESLAM (Mahdi Johari et al., 2022) | 2.5 | **1.1** | 2.4 | 2.0 |
| Point-SLAM (Sandström et al., 2023a) | 4.3 | 1.3 | 3.5 | 3.0 |
| SplaTAM (Keetha et al., 2023) | 3.4 | 1.2 | 5.2 | 3.3 |
| MonoGS (Matsuki et al., 2023b) | **1.6** | 1.4 | **1.5** | **1.5** |
| Gaussian SLAM (ours) | 2.6 | 1.3 | 4.6 | 2.9 |

Table 7: **Experiment on apartment0.** ✗ indicates the methods failed on this large scene.

| | ESLAM | SplaTAM | MonoGS | Ours |
|---|---|---|---|---|
| ATE ↓[cm] | ✗ | ✗ | ✗ | **2.28** |

Table 8: **Tracking performance on ScanNet dataset** (ATE RMSE↓ [cm]). Tracking on ScanNet is especially challenging due to low-quality depth maps and motion blur.

| Method | 0000 | 0059 | 0106 | 0169 | 0181 | 0207 | Avg. |
|---|---|---|---|---|---|---|---|
| NICE-SLAM (Zhu et al., 2022) | 12.00 | 14.00 | 7.90 | 10.90 | 13.40 | **6.20** | 10.70 |
| Vox-Fusion (Yang et al., 2022a) | 68.84 | 24.18 | 8.41 | 27.28 | 23.30 | 9.41 | 26.90 |
| ESLAM (Mahdi Johari et al., 2022) | **7.3** | 8.5 | 7.5 | 6.5 | 9.0 | 5.7 | **7.4** |
| Point-SLAM (Sandström et al., 2023a) | 10.24 | **7.81** | 8.65 | 22.16 | 14.77 | 9.54 | 12.19 |
| SplaTAM (Keetha et al., 2023) | 12.83 | 10.10 | 17.72 | 12.08 | 11.10 | 7.46 | 11.88 |
| MonoGS (Matsuki et al., 2023b) | 9.8 | 32.1 | 8.9 | 10.7 | 21.8 | 7.9 | 15.2 |
| Gaussian SLAM (ours) | 24.75 | 8.63 | 11.27 | 14.59 | 18.70 | 14.36 | 15.38 |

Table 9: **Tracking performance on ScanNet++ dataset** (ATE RMSE ↓ [cm]). Our tracking proves to be robust and competitive in various real-world scenes.

| Method | b20a261fdf | 8b5caf3398 | fb05e13ad1 | 2e74812d00 | 281bc17764 | Avg. |
|---|---|---|---|---|---|---|
| Point-SLAM (Sandström et al., 2023a) | 246.16 | 632.99 | 830.79 | 271.42 | 574.86 | 511.24 |
| ESLAM (Mahdi Johari et al., 2022) | 25.15 | 2.15 | 27.02 | 20.89 | 35.47 | 22.14 |
| SplaTAM (Keetha et al., 2023) | 1.50 | **0.57** | **0.31** | 443.10 | 1.58 | 89.41 |
| MonoGS (Matsuki et al., 2023b) | 7.00 | 3.66 | 6.37 | 3.28 | 44.09 | 12.88 |
| Gaussian SLAM (ours) | **1.37** | 5.97 | 2.70 | **2.35** | **1.02** | **2.68** |

Our method performs on par with other existing dense SLAM methods. Figure 4 shows our successful reconstructions of the large apartment scenes from the Replica dataset, whereas the top-performing baselines—ESLAM, SplaTAM, and MonoGS—fail in tracking, *c.f.* Table 7, and are therefore unable to reconstruct the complete scenes.

Table 10: **Reconstruction performance on Replica dataset.** Our method is comparable to the SOTA baseline Point-SLAM (Sandström et al., 2023a) which requires ground truth depth maps for inference while superior to other dense SLAM methods.

| Method | Metric | Rm0 | Rm1 | Rm2 | Off0 | Off1 | Off2 | Off3 | Off4 | Avg. |
|---|---|---|---|---|---|---|---|---|---|---|
| NICE-SLAM (Zhu et al., 2022) | Depth L1 [cm]↓ | 1.81 | 1.44 | 2.04 | 1.39 | 1.76 | 8.33 | 4.99 | 2.01 | 2.97 |
| | F1 [%]↑ | 45.0 | 44.8 | 43.6 | 50.0 | 51.9 | 39.2 | 39.9 | 36.5 | 43.9 |
| Vox-Fusion (Yang et al., 2022a) | Depth L1 [cm]↓ | 1.09 | 1.90 | 2.21 | 2.32 | 3.40 | 4.19 | 2.96 | 1.61 | 2.46 |
| | F1 [%]↑ | 69.9 | 34.4 | 59.7 | 46.5 | 40.8 | 51.0 | 64.6 | 50.7 | 52.2 |
| ESLAM (Mahdi Johari et al., 2022) | Depth L1 [cm] ↓ | 0.97 | 1.07 | 1.28 | 0.86 | 1.26 | 1.71 | 1.43 | 1.06 | 1.18 |
| | F1 [%] ↑ | 81.0 | 82.2 | 83.9 | 78.4 | 75.5 | 77.1 | 75.5 | 79.1 | 79.1 |
| Point-SLAM (Sandström et al., 2023a) | Depth L1 [cm]↓ | 0.53 | **0.22** | **0.46** | **0.30** | 0.57 | **0.49** | **0.51** | 0.46 | **0.44** |
| | F1 [%]↑ | 86.9 | **92.3** | **90.8** | **93.8** | **91.6** | **89.0** | **88.2** | 85.6 | **89.8** |
| SplaTAM (Keetha et al., 2023) | Depth L1 [cm]↓ | **0.43** | 0.38 | 0.54 | 0.44 | 0.66 | 1.05 | 1.60 | 0.68 | 0.72 |
| | F1 [%]↑ | **89.3** | 88.2 | 88.0 | 91.7 | 90.0 | 85.1 | 77.1 | 80.1 | 86.1 |
| MonoGS (Matsuki et al., 2023a) | Depth L1 [cm]↓ | 3.30 | 3.76 | 4.89 | 2.83 | 6.97 | 6.47 | 5.55 | 4.66 | 4.80 |
| | F1 [%]↑ | 35.0 | 26.0 | 26.0 | 34.0 | 17.0 | 24.0 | 26.0 | 27.0 | 26.88 |
| Gaussian SLAM (ours) | Depth L1 [cm]↓ | 0.61 | 0.25 | 0.54 | 0.50 | **0.52** | 0.98 | 1.63 | **0.42** | 0.68 |
| | F1 [%]↑ | 88.8 | 91.4 | 90.5 | 91.7 | 90.1 | 87.3 | 84.2 | **87.4** | 88.9 |

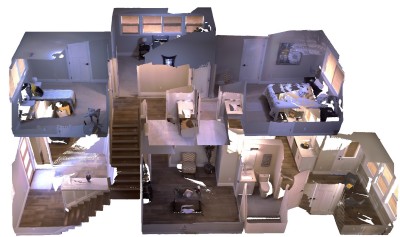

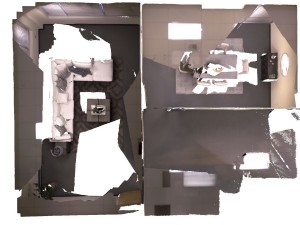

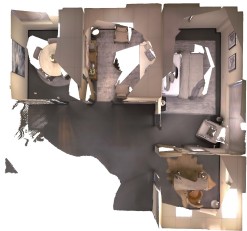

(a) Apartment 0          (b) Apartment 1          (c) Apartment 2

Figure 4: **Qualitative reconstruction of ours on Replica apartment scenes.** 3DGS-based methods SplaTAM and MonoGS, as well as the SOTA method ESLAM failed completing the apartment scenes.

**Ablation Study.** We ablate the impact of the soft alpha mask ($M_{\text{alpha}}$) and the inlier mask ($M_{\text{inlier}}$) on tracking performance, as well as the effectiveness of isotropic regularization on Gaussian splats. Additionally, we justify our decision not to use spherical harmonics for tracking. The details are provided in the supplementary.

**Runtime and Memory Analysis.** In Table 11 we compare runtime and memory usage on the Replica `office0` scene. We report both per-iteration and total runtime. The per-iteration runtime is calculated as the optimization time spent on one frame divided by the iterations, while the total runtime measures all frames.

Table 11: **Runtime and Memory Analysis on Replica `office0`.** Per-iteration runtime is calculated as the optimization time spent on one frame divided by the iterations. The total runtime measures optimization time on all frames. All metrics are profiled using an NVIDIA RTX A6000 GPU.

| Method | Mapping /Iteration(ms) | Mapping Total(m) | Tracking /Iteration(ms) | Tracking Total(m) | Rendering (FPS) | Peak GPU Use(GiB) |
|---|---|---|---|---|---|---|
| NICE-SLAM (Zhu et al., 2022) | 89 | 38.3 | 27 | 35.3 | 2.64 | 12.0 |
| Vox-Fusion (Yang et al., 2022a) | 98 | 49.0 | 64 | 64.0 | 1.63 | 17.6 |
| ESLAM (Mahdi Johari et al., 2022) | 30 | **20.7** | 18 | **5.0** | 0.65 | 17.1 |
| Point-SLAM (Sandström et al., 2023a) | 57 | 117.3 | 27 | 37.0 | 2.96 | 7.7 |
| SplaTAM (Keetha et al., 2023) | 81 | 163.0 | 67 | 90.0 | **2175** | 18.5 |
| Gaussian-SLAM (ours) | **24** | 31.0 | **14** | 27.7 | **2175** | **4.2** |

**Limitations and Future Work.** Although we have effectively used 3D Gaussians for online dense SLAM, tracking a camera trajectory on data with lots of motion blur and low-quality depth maps remains challenging. The trajectory drift is inevitable in frame-to-model tracking without additional techniques like loop-closure or bundle adjustment which might be an interesting future work. Finally, while the sub-mapping strategy is effective in reducing video memory consumption, it introduces redundancy when storing and retrieving sub-maps from the disk.

## 5 Conclusion

We introduced Gaussian-SLAM, a dense SLAM system based on 3D Gaussian Splatting as the scene representation that enables unprecedented re-rendering capabilities. We proposed effective strategies for efficient seeding and online optimization of 3D Gaussians, their organization in sub-maps for better scalability, and a frame-to-model tracking algorithm. Compared to previous SOTA neural SLAM systems like Point-SLAM (Sandström et al., 2023a) we achieve faster tracking and mapping while obtaining better rendering results on synthetic and real-world datasets. We demonstrated that Gaussian-SLAM yields top results in rendering, camera pose estimation, and scene reconstruction on a variety of datasets.

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
