# OpenReview forum: "Gaussian-SLAM: Photo-realistic Dense SLAM with Gaussian Splatting"
_TMLR — Rejected by TMLR_

### Review · Reviewer_GTKS · 2024-08-27

**Summary Of Contributions:**

- The paper introduces an RGB-D SLAM method based on 3D Gaussian Splatting.
- It proposes a submapping framework for 3D Gaussian clouds.
- The method demonstrates increased performance on certain benchmark metrics.

**Audience:**

No

**Broader Impact Concerns:**

I don't see any broader impact concerns.

**Claims And Evidence:**

No

**Requested Changes:**

The requested changes align with the weaknesses noted:

- Include comparisons with prior works (Co-SLAM, GS-SLAM, MonoGS, Photo-SLAM, GS-ICP SLAM, CG-SLAM) in all evaluation tables to clearly demonstrate the effectiveness of the proposed method.
- Discuss the potential risks of submapping and clarify how these risks are addressed.
- Explain the inconsistency between the results in Table 6 and Table 3.
- Add the total runtime to Table 9.
- Remove the claim of concurrency regarding other 3DGS SLAM works.
- Clarify the issues in Equations (13) and (11), and explanation with pose derivatives.

**Strengths And Weaknesses:**

### Strengths:

- **3D Gaussian Submapping**: The 3D Gaussian submapping approach is new within Gaussian-based SLAM, although it is an idea that has already been well-explored in similar point cloud and surfel-based SLAM methods.
- **Performance Benchmarks**: The benchmarks included in the paper highlight the increased performance of the proposed method.

### Weaknesses:

- **Potential Complications of Submapping**: Submapping could complicate the SLAM system. The method creates new submaps based on predefined distances or angles, which may lead to unnecessary submap creation even in previously visited regions. For instance, what happens if the camera repeatedly moves left and right within the same space but keep creating new submap?
- **Insufficient Evaluation**: The experimental results lack comparisons with several seminal baseline methods such as Co-SLAM, GS-SLAM, MonoGS, Photo-SLAM, GS-ICP SLAM, and CG-SLAM, all of which have been accepted to computer vision venues prior to this TMLR review. Without including these baselines in all tables (either by reporting their published results or re-running them using open-source code if available), the evaluation is not sufficiently convincing.
- **Limited Improvement in Real-World Datasets**: Even without the aforementioned baselines, the method does not show clear improvement in real-world datasets like TUM and ScanNet, raising questions about the effectiveness of the proposed submapping framework.
- **Inconsistency in Results**: In Table 6, the proposed method shows poor camera pose estimation on ScanNet, yet in Table 3 it shows superior rendering performance on the same sequences. This discrepancy suggests that the Gaussians might be overfitted to the training frames, and raises the quention of the value of the rendering performance evaluation.
- **Unclear Runtime Measurement**: Table 9 does not provide the total amount of time required to process the sequence, which is a critical concern for a real-time SLAM system. While the table offers a component analysis of tracking and mapping, the definitions of each module are unclear. The total runtime needs to be included alongside the component analysis to give a complete picture of the system's performance.
- **Incorrect Concurrent Work Claims**: The paper claims that GS-SLAM, SplaTAM, MonoGS, and Photo-SLAM are concurrent works. However, this is not true because these works were accepted well before this TMLR review period. The claim need to be corrected.
- **Confusing Equations**: Equation (13) is confusing as it defines camera pose T, which uses the same notation as transmittance in Equation (3). Additionally, the summation of quaternion vectors in Equation (13) is unclear. Further clarification is needed.
- **Isotropic Regularization**: The difference between the isotropic regularization in Equation (11) and the one proposed in MonoGS is uncllear.
- **Implementation Details**: The paper states, "our tracking does not rely on explicitly computed camera pose derivatives and is implemented in PyTorch." However, the benefits of implementing pose derivatives in PyTorch are not explained. Given that all Gaussian derivatives and depth gradients are implemented in CUDA, it is unclear why only the camera pose is implemented differently.

---

> ### Author Response · Authors · 2024-09-07
> **Reply to Review of Paper3028 by Reviewer GTKS**
>
> We sincerely thank the reviewer for their valuable feedback and insightful comments, which have helped us improve the quality of our submission. Here, we address some of the questions and concerns raised by the reviewer.
>
>
> **Incorrect Concurrent Work Claims:** It is true that by the time of the TMLR submission works such as  GS-SLAM, SplaTAM, MonoGS, and Photo-SLAM are already in the proceedings and removed the concurrency claims from the paper.
> However, we have to point out that our work was still one of the first in the field, was released concurrently, and already has several follow-ups.
>
>
> **Insufficient Evaluation:** ESLAM, SplaTAM, and MonoGS (https://arxiv.org/abs/2402.13255) are superior to CoSLAM, GS-SLAM, and PhotoSLAM in tracking, mapping, and efficiency. Therefore, we included additional results for SplaTAM and MonoGS in the paper. GS-ICP SLAM and CG-SLAM have not yet been published in the proceedings.
>
>
> **Potential Complications of Submapping:** The downside of sub-mapping is potential map redundancies when they are stored on the disk. However, in practice, we did not experience any problems on any dataset. We sacrifice disk memory for computation speed and VRAM. We added this to the limitations section of the paper.
>
> **Limited Improvement in Real-World Datasets:** Our method outperforms SplaTAM by 8dB on average PSNR on both ScanNet and Replica datasets which we believe is a substantial improvement.
>
>
> **Inconsistency in Results:** For TUM-RGBD and ScanNet datasets there are no hold-out test views on which one can test the novel view synthesis performance. Therefore, the method’s performance is tested on the training views similar to other baselines. To close this gap, we added novel view synthesis evaluation on ScanNet++ where test views are available. Initially, we included this table in the supplementary material of the submission. Realizing that this is an important evaluation we now added it to the main body of the paper and extended it with more methods.
>
>
> **Unclear Runtime Measurement:** Thanks for the suggestion, we added the total runtime to Table 9.
>
>
> **Confusing Equations:** Thanks for spotting the quaternion typo in Equation (13). The pose initialization notation is corrected in the paper.
>
>
> **Isotropic regularization:** MonoGS applies a similar soft constraint on isotropy, with the difference that we regularize with the mean size of the submap instead of the global map.
>
>
> **Implementation Details:** The authors of MonoGS analytically compute the derivatives and implement their computation directly on CUDA. Our method implicitly optimizes the camera’s rotation and translation by rotating and translating the submap to minimize rendering color and depth losses in PyTorch. While explicit camera gradient computation is faster, it requires additional camera optimization steps during the mapping stage over a set of keyframes covering a large area. This, in turn, requires holding the whole 3D Gaussian map in VRAM preventing the method from scaling to larger scenes and consuming significantly more memory.

---

> > ### Comment · Reviewer_GTKS · 2024-10-08
> >
> > Thank you for your response. I have a few more questions and comments:
> >
> > - What is the total size of the reconstructed map, in terms of megabytes or gigabytes? I believe this should be discussed in addition to peak GPU usage, as it is common to evaluate the total map size in dense neural SLAM systems. My assumption is that naive submapping could result in redundant maps.
> > - For novel view synthesis, how do you choose the appropriate submap to render when there are multiple submaps covering the same area?
> > - Regarding the rendering performance evaluation, the paper follows the evaluation pipeline proposed in Point-SLAM. According to Point-SLAM paper, the rendering metrics are evaluated on every 5th input frame. However, in the proposed method, every 5th frame is used as a keyframe (training view). Such training-view rendering does not make sense as a performance evaluation.
> > - For runtime measurement, I expect the total runtime of the entire SLAM system (i.e., the total time required to process all input frames), rather than just individual tracking or mapping modules.

---

> > > ### Author Response · Authors · 2024-10-11
> > > **Reply to Review of Paper3028 by Reviewer GTKS**
> > >
> > > **Novel views rendering**. To render novel views, we merge the sub-maps into a single map. For more details, we refer to section 2 *further implementation details* in the supplementary material.
> > >
> > >
> > > **Total map size**.  You are right, sub-mapping indeed results in redundancy in disk space. This is a deliberate design choice assuming that disk space is cheaper than GPU memory. As shown in Table 11 of the main paper, this results in significantly lower peak GPU memory usage for our method. For instance, on the Replica office0 scene (22 m^2 in size), ESLAM and SplaTAM require 17.1 GB and 18.5 GB of VRAM respectively, while our method consumes at most 4.2 GB of VRAM.
> > >
> > > In terms of map size, the total size of the sub-maps on the office0 scene adds up to 2.4 GB. The size of the merged Gaussian map is 49.52 MB. For comparison, the Gaussian map size on this scene of SplaTAM is 404.5 MB.
> > >
> > > **Rendering evaluation**. It is a common practice in Neural SLAM to evaluate how well a method can re-render the training views. For example, in NICE-SLAM, Point-SLAM, ESLAM, etc., every fifth frame was used as a keyframe. We follow the accepted evaluation protocol to be comparable to previous methods. We acknowledge that this form of evaluation has its limitations, as methods can overfit the training views. To make the evaluation of our method more comprehensive, in Table 4, we added novel-view synthesis evaluation on the ScanNet++ dataset with an official train/test frames split. Our method shows competitive performance in both training and novel view synthesis.
> > >
> > > **Total Runtime**. Typically only tracking and mapping runtimes are reported since there are many things in a pipeline that slow it down such as checkpointing, logging, and visualizations. The total runtime of our method on the Replica office0 scene (2000 frames) is 73 minutes with an NVIDIA RTX A6000 GPU. For comparison, the total runtime of SplaTAM in this scene is 285 minutes.

---

> > > > ### Author Response · Authors · 2024-11-21
> > > > **Reply to Review of Paper3028 by Reviewer GTKS**
> > > >
> > > > We hope this message finds you well. We are writing to follow up on the responses we provided to your review. We understand that you have many commitments, but we would appreciate any further feedback you might have. If there are additional questions or clarifications needed, please let us know, and we will be glad to assist. Thank you for your time and consideration.

---

### Review · Reviewer_jEJT · 2024-09-04

**Summary Of Contributions:**

The paper introduces a dense SLAM system using 3D Gaussians for scene representation. A key contribution is the use of sub-maps to manage GPU memory, prevent overfitting, and improve scalability during mapping.

**Audience:**

Yes

**Broader Impact Concerns:**

Not applied.

**Claims And Evidence:**

Yes

**Requested Changes:**

Clarity
- The sub-map creation method relies solely on spatial thresholds (camera rotation and translation). However, rapid camera movements or accelerations could justify earlier sub-map initialization, even if the spatial thresholds aren't met
- Uniformly sampling Gaussians in high color gradient regions is useful for capturing scene details but may introduce bias toward visually prominent areas, potentially overlooking important geometric features in low gradient regions
- In the implementation details, the meaning of thresholds “Mu” and “Mc” is unclear. Though mentioned in the sub-map building section, it would help to restate or reference where these variables are defined for better clarity


Typos/formatting issues
The paper contains several typos and formatting issues that disrupt readability. Notable examples include:
- In the methods section, "Figure" is missing before "Figure 2" in the phrase "3D map representation. Figure 2 provides an overview."
- Under sub-map optimization, "Table" is missing in "optimization and improve tracking Table 11."
- In the rendering performance section, several instances of the word "Figure" are missing.
- In reconstruction performance, there’s an issue with spacing between "SplaTAMKeetha et al."

**Strengths And Weaknesses:**

Strengths:
- The sub-map creation strategy is novel in this context, effectively reducing GPU memory usage and enhancing scalability.
- The evaluations are thorough, covering both synthetic and real-world datasets.

Weakness
- The technical novelty of the method is limited, with the system's overall contribution being more significant.
- While sub-maps reduce GPU memory usage, the need to retrieve them may introduce GPU-to-CPU overhead during loop closures for long term mapping. It would be beneficial to discuss this trade-off.
- A comparison with more recent RGBD SLAM systems, such as MonoGS, would strengthen the evaluation, beyond the SH ablation in Table 12
- The system’s tracking performance is sensitive to motion blur and low-quality depth maps, common in real-world scenarios. It would be helpful if the authors discussed potential improvements to tracking under these conditions.
- How valid is constant speed assumption any analysis on the converge basin of the proposed method?

---

> ### Author Response · Authors · 2024-09-07
> **Reply to Review of Paper3028 by Reviewer jEJT**
>
> We would like to express our sincere gratitude to the reviewer for their valuable feedback and insightful comments, which have improved the quality of our submission. Below, we address the questions and concerns raised in the review.
>
>
> **Sub-maps reduce GPU memory:** Our method doesn’t require sub-map retrieval during the runtime which removes the computational overhead required to move the sub-maps from CPU and GPU several times.
>
>
> **Comparison with more recent RGBD SLAM systems:** We added more baselines to the tracking, rendering, and reconstruction tables.
>
> **Motion blur sensitivity:** A potential improvement for motion blur sensitivity is a stronger camera pose initialization. This would allow for avoiding local minima on challenging sequences like ScanNet.
>
>
> **Constant speed assumption:** Constant speed assumption performance is very scene-dependent. For some datasets like KITTI, it is a very useful assumption while for datasets captured with a hand-held device, it doesn’t work that well. Moreover, we use constant speed assumption to be comparable with other methods like Point-SLAM, SplaTAM, NICE-SLAM, Vox-Fusion, etc.
>
>
> **The sub-map creation method:** New sub-maps are created only when camera rotation or translation exceeds the thresholds.
>
> **Uniformly sampling Gaussians:** We sampled more 3D Gaussians in high-gradient regions because they were harder to represent with 3D Gaussians. Low-gradient regions can be reconstructed with fewer 3D Gaussians of a larger size as mentioned in the original GaussianSplatting paper.
>
>
> **Implementation details:** We clarified the meaning of Mu and Mc in the implementation details section.
>
> **Formatting issues:** Thanks for spotting the typos and formatting issues. We addressed them in the newly uploaded paper.

---

> > ### Author Response · Authors · 2024-11-21
> > **Reply to Review of Paper3028 by Reviewer jEJT**
> >
> > We hope this message finds you well. We are writing to follow up on the responses we provided to your review. We understand that you have many commitments, but we would appreciate any further feedback you might have. If there are additional questions or clarifications needed, please let us know, and we will be glad to assist. Thank you for your time and consideration.

---

### Review · Reviewer_M2uo · 2024-09-29

**Summary Of Contributions:**

The authors propose “Gaussian-SLAM”, a simultaneous localization and mapping (SLAM) algorithm based on 3D Gaussian Splitting (3DGS) as mapping representation in the RGBD setting. While the main framework is similar to previous works, the main contribution of the paper is the proposal of many practical techniques applied at different stages to the original 3DGS specifically tailored for the SLAM purposes.
The results in the experiments section show that Gaussian-SLAM achieves best rendering quality but mixed performance in terms of tracking and reconstruction against other neural representation-based SLAM algorithms, but not with traditional feature-based baselines like ORB-SLAM. While it can be informative to obtain high-quality visual renderings during the SLAM process, the more important metric in the SLAM process to me is its tracking performance, which has been a bottleneck for neural-representation-based SLAM.

**Audience:**

Yes

**Broader Impact Concerns:**

I do not observe any direct ethical concerns of the method, except for the potential privacy concerns of confidential or unauthorized visual information during the SLAM process. But as far as this paper is concerned, the datasets it uses are publicly available and legal to use.

**Claims And Evidence:**

Yes

**Requested Changes:**

1. Perform ablation studies to verify the efficacy of design changes.
2. Compare to feature-based SLAM methods in terms of tracking performance.
3. Fix the missing table reference in section sub-map optimization.
4. Add reference to distributed NeRFs in related works.

**Strengths And Weaknesses:**

Strengths:
1. The automatic sub-map chunking/initialization based on their deviation of the initial frame in terms of either translation or rotation difference is a practical solution with good intuition.
2. The sub-map building process is modified quite extensively to fit the SLAM purposes. The key change is that it does not perform the “add and prune” procedure commonly used in 3DGS optimization, but instead focus on using a procedure to ensure that the geometric density of the Gaussians are close to the ones obtained from the depth sensor. Such design does have its advantage for speed-up, training stability and potentially more robust tracking performance.
3. Thanks to the design in sub-map building, in sub-map optimization, it can speed up by depth gradient calculation caching two components in the partial derivatives that stay constant during the optimization procedure. This is not the case if the traditional 3DGS optimization pipeline is applied.
4. The visual rendering quality is the best against other neural-rendering-based SLAM methods across different benchmarks.

Weaknesses:
1. The paper introduces many practical modifications to the traditional 3DGS pipeline for the SLAM purposes that are intuitively helpful. However, such big changes also bring the worry of deviation of the 3DGS best practices. The paper lacks proper ablation study to verify the efficacy of its design change.
2. While the rendering quality of the method is good, it does not translate to better tracking and reconstruction performance. In fact, it does not compare to the feature-based SLAM methods like ORB-SLAM, which usually has superior tracking performance than neural-representation-based SLAM methods.
3. The paper seems to miss a line of research in the related works about distributed NeRFs including Block-NeRF, NeRFuser, NeRFusion.

---

> ### Author Response · Authors · 2024-10-11
> **Reply to Review of Paper3028 by Reviewer M2uo**
>
> 1. We ablate over usage of spherical harmonics in the supplement Table 2, and the use of isotropic regularization in the supplement Table 3 and 4. Please let us know if you are interested in any additional experiments.
>
> 2. We provide the average ATE of ORB-SLAM3 on all our datasets. Our method shows superior performance when provided with high-quality color and depth maps (Replica and ScanNet++ datasets). Furthermore, ORB-SLAM3 can not deal with the large interframe motion as seen in ScanNet++ scenes.
>
> | Method/Dataset | Replica | TUM_RGBD | ScanNet | ScanNet++ |
> |----------------|---------|----------|---------|-----------|
> | ORB-SLAM3       | 1.83    | 1.7      | 7.28    | 159.7     |
> | Ours           | 0.31    | 2.9      | 15.38   | 2.68      |
>
> 3. Thanks for spotting the typo. We’ve integrated the fix in the paper.
>
> 4. Thanks for spotting missing references. We’ve added them to the paper.

---

> > ### Author Response · Authors · 2024-11-21
> > **Reply to Review of Paper3028 by Reviewer M2uo**
> >
> > We hope this message finds you well. We are writing to follow up on the responses we provided to your review. We understand that you have many commitments, but we would appreciate any further feedback you might have. If there are additional questions or clarifications needed, please let us know, and we will be glad to assist. Thank you for your time and consideration.

---

> ### Comment · Reviewer_M2uo · 2024-11-27
> **Better tracking performance against ORB-SLAM3 on ScanNet++**
>
> Thanks for the additional experiments and fixes. The experiment against ORB-SLAM is particularly interesting. Although in TUM and ScanNet, ORB-SLAM3 still holds an edge, the big performance edge on ScanNet++ shows potential for 3DGS-based SLAM methods. I would like to see more analysis towards the performance difference on ScanNet++, a relatively new dataset that is comparatively less studied, but such result is enough for the purpose of this paper. I am generally satisfied with the paper at its current status.

---

### Decision · Action_Editor_65z2 · 2025-01-13

**Recommendation:** Reject

**Comment:**

The reviewers largely agree on the paper's strengths and weaknesses. Among the strengths, the reviewers acknowledged that Gaussian-SLAM is the first submap-based approach to SLAM with 3D Gaussian Splatting. The means by which the method modifies the standard 3DGS procedure to accommodate submap-based SLAM may provide useful insights to practitioners. Additionally, at least two reviewers point out that Gaussian-SLAM outperforms existing neural-based SLAM methods on several standard benchmarks.

As noted above, the reviewers shared similar concerns about the experimental evaluation and the extent to which it demonstrates the advantages of Gaussian-SLAM and the contributions of its approach to 3DGS submapping for SLAM. While Gaussian-SLAM results in better rendering performance, the advantages in terms of tracking and reconstruction accuracy are not evident, particularly for real-world settings where the color and depth inputs will likely not be of high-quality. Meanwhile, the reviewers found that the initial submission lacked comparisons to relevant baselines. The authors made an effort to address these issues during the discussion phase, in large part by including additional experimental results. Two of the three reviewers acknowledged these results as well as the authors' response to the reviewers' initial concerns. However, the AE agrees that further discussion and evaluation is necessary to position Gaussian-SLAM in the context of contemporary neural and non-neural approaches to SLAM, particularly with regard to the role of submapping, which is arguably the paper's primary contribution. Further discussion on this point will help to make it clear what can be learned by researchers interested in neural-based approaches to SLAM.

**Audience:**

The focus of the paper---a submap-based approach to SLAM using 3D Gaussian Splatting as the environment representation---is relevant to the subset of the TMLR community focused on robot mapping and localization, and 3D vision.

**Claims And Evidence:**

The paper proposes Gaussian-SLAM, an RGBD-based simultaneous localization and mapping (SLAM) framework that represents the map using 3D Gaussian Splatting (3DGS). Integral to Gaussian-SLAM is its representation of the environment map as a collection of smaller submaps that are separately maintained and optimized, which has potential advantages in terms of computational and memory efficiency.

The reviewers raised questions about the experimental evaluation in terms of both the method's performance and the absence of relevant baselines in the paper as initially submitted. In terms of performance, Reviewers N2uo and GTKS point out that while Gaussian-SLAM exhibits better rendering quality relative to the baselines that were considered, the improvement gains do not translate to more accurate tracking and reconstruction, including when compared to traditional feature-based SLAM methods (e.g., ORB-SLAM). As part of the discussion phase, the authors presented additional results that show that Gaussian-SLAM outperforms ORB-SLAM3 on Replica and ScanNet++ in terms of tracking error when provided with higher-quality color and depth inputs. Reviewer GTKS finds that the method does not exhibit clear improvements on real-world datasets like TUM and ScanNet, which as noted by Reviewer jEJT, typically involve lower-quality RGBD images (e.g., due to motion blur). While Gaussian-SLAM does offer advantages in terms of rendering quality, the accuracy of tracking and reconstruction are arguably more important in the context of SLAM. Tracking and reconstruction aside, there were concerns about the fairness of using keyframes to evaluate rendering accuracy. Meanwhile, Reviewers GTKS and jEJT found that the initial submission lacked comparisons to relevant baselines (including MonoGS, Co-SLAM, GS-SLAM, MonoGS, Photo-SLAM, GS-ICP SLAM, and CG-SLAM), which the authors made a concerted effort to address during the discussion phase.

Overall, the primary contribution of the paper is its proposal of a submap-based approach to SLAM with 3DGS representations. Previous work exists on 3DGS-based SLAM as well as on submap-based SLAM for non-3DGS representations, however this is the first paper on 3D Gaussian submapping for SLAM. However, it is not evident what can be learned from the paper in its current form, in part based upon the  the experimental evaluation, inclusive of Gaussian-SLAM's tracking and reconstruction accuracy as well as its computational efficiency, or the description of how submaps are created and maintained.

**Resubmission Of Major Revision:**

The authors may consider submitting a major revision at a later time.